# Sex Chromosome Turnover in Bent-Toed Geckos (*Cyrtodactylus*)

**DOI:** 10.3390/genes12010116

**Published:** 2021-01-19

**Authors:** Shannon E. Keating, Madison Blumer, L. Lee Grismer, Aung Lin, Stuart V. Nielsen, Myint Kyaw Thura, Perry L. Wood, Evan S. H. Quah, Tony Gamble

**Affiliations:** 1Department of Biological Sciences, Marquette University, Milwaukee, WI 53233, USA; stuart.nielsen@ufl.edu (S.V.N.); tgamble@geckoevolution.org (T.G.); 2Keck Science Department, Scripps College, Claremont, CA 91711, USA; mraemb@gmail.com; 3Herpetology Laboratory, Department of Biology, La Sierra University, Riverside, CA 92515, USA; lgrismer@gmail.com; 4Fauna and Flora International, No (35), 3rd Floor, Shan Gone Condo, Myay Ni Gone Market Street, Sanchaung Township, Yangon 11111, Myanmar; aung.lin@fauna-flora.org; 5Department of Natural Sciences, University of Michigan-Dearborn, Dearborn, MI 48128, USA; 6Department of Herpetology, Florida Museum of Natural History, Gainesville, FL 31611, USA; 7Myanmar Environment Sustainable Conservation, Yangon 11181, Myanmar; mgmyint.banca@gmail.com; 8Department of Biological Sciences and Museum of Natural History, Auburn University, Auburn, AL 36849, USA; perryleewoodjr@gmail.com; 9Institute of Tropical Biodiversity and Sustainable Development, University Malaysia Terengganu, Kuala Nerus, Terengganu 21030, Malaysia; evanquah@umt.edu.my; 10Milwaukee Public Museum, 800 W. Wells St., Milwaukee, WI 53233, USA; 11Bell Museum of Natural History, University of Minnesota, 2088 Larpenteur Ave. W., St. Paul, MN 55113, USA

**Keywords:** Gekkota, lizard, RADseq, synteny

## Abstract

Lizards and snakes (squamates) are known for their varied sex determining systems, and gecko lizards are especially diverse, having evolved sex chromosomes independently multiple times. While sex chromosomes frequently turnover among gecko genera, intrageneric turnovers are known only from *Gekko* and *Hemidactylus*. Here, we used RADseq to identify sex-specific markers in two species of Burmese bent-toed geckos. We uncovered XX/XY sex chromosomes in *Cyrtodactylus chaunghanakwaensis* and ZZ/ZW sex chromosomes in *Cyrtodactylus pharbaungensis*. This is the third instance of intrageneric turnover of sex chromosomes in geckos. Additionally, *Cyrtodactylus* are closely related to another genus with intrageneric turnover, *Hemidactylus*. Together, these data suggest that sex chromosome turnover may be common in this clade, setting them apart as exceptionally diverse in a group already known for diverse sex determination systems.

## 1. Introduction

Squamate reptiles (lizards and snakes) have evolved a number of different sex-determining systems [1,2,3,4,5,6,7,8], but gekkotan lizards stand out in terms of their diversity. The ancestor of geckos is hypothesized to have temperature-dependent sex determination (TSD) followed by repeated transitions to both XX/XY and ZZ/ZW sex chromosomes with some species retaining the ancestral TSD [3,5]. Gecko sex determination is so labile that there are even two gecko genera that exhibit intrageneric transitions. *Hemidactylus* and *Gekko* include species with both XX/XY and ZZ/ZW sex chromosomes [5,9,10,11]. The repeated sex chromosome turnovers in geckos provide opportunities for focused study on the evolutionary forces driving sex chromosome transitions and their early evolution. By identifying additional turnover events, we increase our power to investigate these fundamental evolutionary questions.

Bent-toed geckos (*Cyrtodactylus*) are the most species rich gecko genus with >300 described species distributed throughout South and Southeast Asia and Oceania [12,13,14,15,16,17]. Despite this diversity, we only know the sex-determining system of one species, the Bornean endemic *Cyrtodactylus pubisulcus*, whose ZZ/ZW sex chromosomes were identified through traditional cytogenetics [18]. Other *Cyrtodactylus* karyotypes have failed to reveal heteromorphic, or visually distinct, sex chromosomes [18,19]. Nonetheless, investigating *Cyrtodactylus* sex chromosomes is a worthwhile endeavor because *Cyrtodactylus* is closely related to the aforementioned *Hemidactylus*, a genus with both XX/XY and ZZ/ZW species [12,20]. It stands to reason that the plasticity observed within *Hemidactylus* may extend to *Cyrtodactylus*.

Here, we investigated sex chromosome systems in two species of bent-toed geckos from Myanmar. Using restriction-site-associated DNA sequencing (RADseq [21]), we found that *Cyrtodactylus pharbaungensis* possesses a ZZ/ZW sex chromosome system, while *Cyrtodactylus chaunghanakwaensis* has an XX/XY system. When considered in light of the previously recognized ZZ/ZW system in *C. pubisulcus* [18], these results indicate at least one turnover has occurred within *Cyrtodactylus*. This discovery further highlights the extraordinary number of transitions among gekkotan sex chromosomes and the enormous gap in our knowledge surrounding the phylogenetic distribution of gecko sex chromosomes and evolutionary processes that lead to sex chromosome transitions.

## 2. Materials and Methods

*Cyrtodactylus chaunghanakwaensis* [22] and *C. pharbaungensis* [23] are range-restricted endemics known only from small isolated karstic hills from Mon State in the Salween Basin of southern Myanmar, and were collected from their type localities at Chaunghanakwa Hill and Pharbaung Cave (Appendix A). We extracted DNA from eight females and eleven males of *C. pharbaungensis* and ten females and nine males of *C. chaunghanakwaensis* using the QIAGEN DNeasy Blood and Tissue Kit. We prepared RADseq libraries using a modified protocol from Etter et al. [5,24]. Briefly, we digested genomic DNA using a high-fidelity *Sbf1* restriction enzyme (New England Biolabs, Ipswich, MA, USA), and ligated individually barcoded P1 adapters to each sample. We pooled samples into multiple libraries, sonicated, and size selected for 200–500 bp fragments using magnetic beads in a PEG/NaCl buffer [25]. We then blunt-end repaired, dA-tailed, and ligated pooled libraries with P2 adapters containing unique Illumina i7 (San Diego, CA, USA) indices. We amplified pooled libraries using NEBNext Ultra II Q5 polymerase (New England Biolabs) for 16 cycles and size selected a second time for 250–650 bp fragments that now contained full Illumina adapters. Libraries were sequenced using paired-end 150-bp reads on an Illumina HiSeq X (San Diego, CA, USA) at Novogene. Cleaned RADseq reads have been deposited to NCBI SRA: SAMN17316796-SAMN17316830.

We analyzed the RADseq data using a previously described bioinformatic pipeline [5]. We demultiplexed, trimmed, and filtered raw Illumina reads using the process_radtags function in STACKS [26] (1.41). We used RADtools [27] (1.2.4) to generate RADtags for each individual and identified candidate loci and alleles from the forward reads. We then used a custom Python script [5] to identify putative sex-specific markers from the RADtools output, i.e., markers found in one sex but not the other. The script also generates a list of “confirmed” sex-specific RAD markers that excludes any sex-specific markers found in the original read files of the opposite sex, thus eliminating false positives. Finally, we used Geneious [28] (R11) to assemble the forward and reverse reads of “confirmed” sex-specific RAD markers. These loci should correspond to genomic regions unique to a single sex, the Y or W chromosome, such that female-specific markers denote a ZZ/ZW system while male-specific markers indicate an XX/XY system.

We validated a subset of the “confirmed” sex-specific markers with PCR. DNA primers (Table 1) were designed using Primer3 [29] in Geneious (R11). We tested ten markers for *C. pharbaungensis* using nine females and ten males and ten markers for *C. chaunghanakwaensis* using ten females and nine males. After PCR, markers were visualized on a 1% agarose gel to determine whether they amplified in a sex-specific pattern.

We queried sex-specified RAD markers against an assembled *Hemidactylus turcicus* transcriptome [30] using BLAST [31] in Geneious (R11). We then used BLAST to compare the matching *H. turcicus* transcripts to genes in the chicken (*Gallus gallus*) genome [32,33], as it is a commonly used point of reference when discussing synteny among amniotes [11,34,35,36,37,38,39]. By identifying the sex chromosome linkage group, we can determine whether *C. pharbaungensis* and *C. chaunghanakwaensis* have sex chromosome systems syntenic with each other or other amniote species. Alternatively, if each species has a different sex chromosome linkage group, then we can conclude they have different sex chromosomes, even if there is no difference in heterogamety.

Identifying sex chromosome turnovers is best done in a phylogenetic framework. Therefore, we generated a phylogeny of *Hemidactylus* and *Cyrtodactylus* species to combine our new results with sex chromosome data from the literature. We accessed ND2 and adjacent tRNAs for 239 *Cyrtodactylus* species, 61 *Hemidactylus* species, and 11 outgroup species from GenBank (Appendix A) and previous publications [16]. We aligned the sequences with Muscle [40] using the default settings in Geneious [28] and generated a maximum likelihood phylogeny using IQ-TREE (v2.1.2) [41]. We first used ModelFinder [42] to select among alternative partition schemes and models of evolution using the Bayesian information criterion (BIC) with four partitions: the first, second, and third codon of ND2, and the associated tRNAs. IQ-TREE then constructed a maximum likelihood tree using the selected models. Nodal support was generated with 1000 ultrafast bootstrap approximations (UFBoot [43]). We used least-square dating (LSD2 [44]) to generate a time tree using secondary calibrations from Gamble et al. [45], utilizing several calibration points between *Hemidactylus* and *Cyrtodactylus* species as well as the outgroups. Confidence intervals were generated with 100 replicates.

All experiments were carried out in accordance with animal use protocols under the Brigham Young University’s Institutional Animal Care and Use Committee (IACUC protocol #160401). Export and collection permits were issued by Mr. Win Naing Thaw of the Ministry of Natural Resources and Environmental Conservation Forest Department.

## 3. Results

RADseq analysis for *C. pharbaungensis* included eleven males and four females—barcode errors resulted in several of the female samples not being used for subsequent analysis. We identified 153,555 RAD loci with two or fewer alleles. Of these, 652 markers were female-specific and none were male-specific. We checked the sex-specific markers against the reads from the opposite sex and retained 646 “confirmed” female-specific RAD markers. We designed PCR primers for ten markers and validated two as sex-specific (Figure 1A). Marker CyrtP_253 produced a small band in females (~400 bp) and a larger band in males (~900 bp). These PCR products correspond to amplification of the W allele in females and the Z allele in males. Although the Z and W alleles have diverged to the point that they are different sizes, enough sequence similarity presumably exists that the primers designed for the W allele still amplify the Z allele in males. In females, the W allele is preferentially amplified over the Z allele, producing a single band. PCR primers designed using the W allele will occasionally produce PCR products from the homogametic sex, although these are typically much fainter than those in the heterogametic sex when visualized on a gel [5,46]. Marker CyrtP_194 amplified strongly in females and not in males. These results indicate *C. pharbaugensis* has a ZZ/ZW sex chromosome system, and the female-specific markers are linked to the W chromosome.

We eliminated one female *C. chaunghanakwaensis*, LSUHC13294, from the RADseq analysis due to low read depth, leaving nine females and nine males. We identified 151,212 RAD markers. We found zero female-specific markers and 168 male-specific markers, of which 166 were “confirmed” by checking them against the original female reads. We designed PCR primers for ten markers and validated three as sex-specific. CyrtC_6, CyrtC_18, and CyrtC_151 all amplified strongly in males but not in females (Figure 1B). These results indicate a XX/XY sex chromosome system where the male-specific RAD markers are localized to the male-specific Y chromosome.

Using *H. turcicus* transcripts as an intermediary step (Appendix A), we identified 38 chicken genes via BLAST of female-specific *C. pharbaungensis* RAD markers. We observed 19 of these genes on chicken chromosome ten. This was significantly more genes than expected (expected = 1; *p* = 5.525818 × 10^−^^21^, Figure 2), assuming the number of expected genes with BLAST hits is proportional to the number of annotated genes on each chromosome. This suggests the ZZ/ZW sex chromosomes of *C. pharbaungensis* are syntenic with chicken chromosome ten. The number of genes with BLAST hits on other chromosomes was equal to or fewer than expected with the exception of chicken chromosome 12, which was not significant (observed = 2, expected = 1; *p* = 0.1938535; Figure 2). *p*-values were calculated using the hypergeometric test in R v.3.6.3 [47,48]. We were unable to confidently identify the sex linkage group for *C. chaunghanakwaensis* as BLAST searches returned only two genes each on different chicken linkage groups (Appendix A).

The ND2 phylogenetic tree (Figure 3A) was largely concordant with previously published phylogenies utilizing similar datasets and methods [16,51,52]. The best fit models of sequence evolution for each partition were: codon one = TVM + F + R7; codon two = TIM + F + R8; codon three = TIM + F + R7; and tRNAs = SYM + R5.

## 4. Discussion

Geckos are known for their diverse sex chromosome systems and frequent turnovers [5,53,54]. This includes sex chromosome transitions among species in at least two gecko genera, *Hemidactylus* and *Gekko* [5,9,10,11]. *Cyrtodactylus* becomes the third gecko genus with an intra-generic transition with the discovery of novel sex chromosome systems in two species of Burmese bent-toed geckos: ZZ/ZW in *C. pharbaungensis* and XX/XY in *C. chaunghanakwaensis*. Despite being a highly diverse genus with >300 species, the sex chromosome systems identified here represent only the second and third known instance of sex chromosomes in the genus. Given the phylogenetic placement of these species and *C. pubisulcus*, a species with ZZ/ZW sex chromosomes, at least one turnover has occurred, possibly more if the ZZ/ZW sex chromosomes of *C. pharbaungensis* and *C. pubisulcus* are not homologous (Figure 3). Hemidactylus, the sister genus to *Cyrtodactylus*, has likewise experienced intrageneric sex chromosome turnovers (Figure 3, [5]), suggesting the *Hemidactylus* + *Cyrtodactylus* clade may be prone to frequent turnover of sex chromosomes.

Because the vast majority of *Cyrtodactylus* species have yet to be studied (Figure 3A) there is almost certainly additional sex chromosome diversity within the genus. *Cyrtodactylus* is the most speciose gecko genus and continues to grow as new species are described every year [55,56]. However, less than 1% of species have a known sex determination system. *Hemidactylus* fares little better, as ~1.8% of described species have a known sex determination system. Further investigations of other *Cyrtodactylus* and *Hemidactylus* species will likely uncover additional sex chromosomes and help to localize transitions onto the phylogeny. Given the diversity of sex chromosome systems observed among *Cyrtodactylus* and *Hemidactylus* (despite the paucity of sampling), these genera form an ideal clade for studying the evolutionary drivers of rapid sex chromosome turnover and the early evolution of sex chromosomes.

Sex chromosomes evolve from autosomes and some autosomes may be more likely to make this change than others [34,57]. However, *C. pharbaungensis* is the first amniote shown to have sex chromosomes syntenic with chicken chromosome ten. Synteny information is vital for identifying sex chromosome turnovers, particularly cis-turnovers where a novel sex chromosome system replaces an ancestral system with the same heterogamety, for example, a novel ZZ/ZW system replacing the ancestral ZZ/ZW system or a novel XX/XY system replacing the ancestral XX/XY system. Since both *C. pharbaungensis* and *C. pubisulcus* have female heterogamety synteny data could allow us to distinguish whether these are the same or different ZZ/ZW systems. Unfortunately, karyotype data used to identify the heteromorphic sex chromosomes of *C. pubisulcus* do not provide any synteny information [18]. If future studies demonstrate that *C. pharbaungensis* and *C. pubisulcus* do not share homologous sex chromosomes, then all three *Cyrtodactylus* species studied thus far have independently derived sex chromosome systems. Alternatively, if *C. pharbaungensis* and *C. pubisulcus* have the same sex linkage group, then at least one turnover occurred in *Cyrtodactylus*, in which an ancestral ZZ/ZW system was maintained in both *C. pharbaungensis* and *C. pubisulcus* but transitioned to an XX/XY system in *C. chaunghanakwaensis*.

Mapping sex-specific *Cyrtodactylus* RAD markers directly to chicken did not yield useful results, but an alternative approach using the *H. turcicus* transcriptome as an intermediate step was successful in *C. pharbaungensis*. Using one or more intermediate steps for BLAST searches has been used previously to identify the sex chromosome linkage group in a frog species with limited genomic resources. In that study, *Rana arvalis* sex-linked markers were iteratively mapped to a *Rana temporaria* draft genome. Those *R. temporaria* fragments were mapped to the *Nanorana parkeri* genome. Finally, the matching *N. parkeri* scaffolds were mapped to the highly contiguous *Xenopus tropicalis* genome [58]. The increasing availability of genomic and transcriptomic data across diverse vertebrate clades should make hierarchical mapping strategies like this a routine part of identifying sex chromosome synteny.

## 5. Conclusions

Here, we uncovered the XX/XY sex chromosomes of *C. chaunghanakwaensis* and the ZZ/ZW sex chromosomes of *C. pharbaungensis*. These results contribute to the ever-increasing diversity of gecko sex chromosome systems. Although squamate lizards are known for a diversity of sex chromosome systems, they tend to have conserved systems within suborders or families, including scincids [59], lacertids [60], varanids [61,62], pleurodonts [63] (but see [6,8]), and caenophidian snakes [36,64]. In contrast, geckos regularly experience sex chromosome turnovers between families and genera, and, as demonstrated here, even within genera. These frequent transitions make geckos a remarkable group within the already remarkable squamates, and highlights their importance as a model for studying sex chromosome evolution.

## Figures and Tables

**Figure 1 genes-12-00116-f001:**
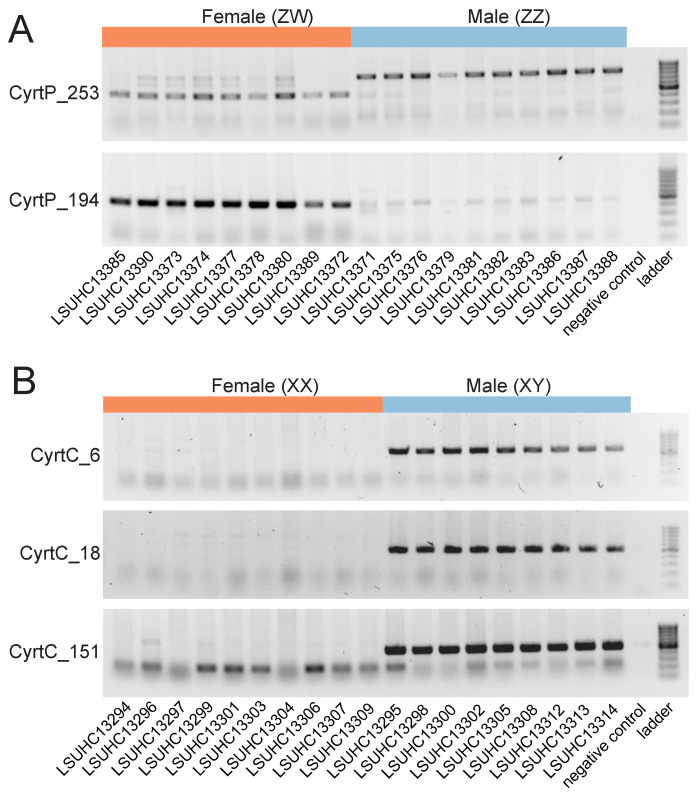
PCR validation of female-specific RAD markers in *C. pharbaugensis* (**A**) and male-specific RAD markers in *C. chaunghanakwaensis* (**B**). Specimen IDs are located beneath gels. Negative control and DNA ladder (Thermo Scientific GeneRuler 100 bp DNA ladder, Waltham, MA, USA) are shown on the right of each gel.

**Figure 2 genes-12-00116-f002:**
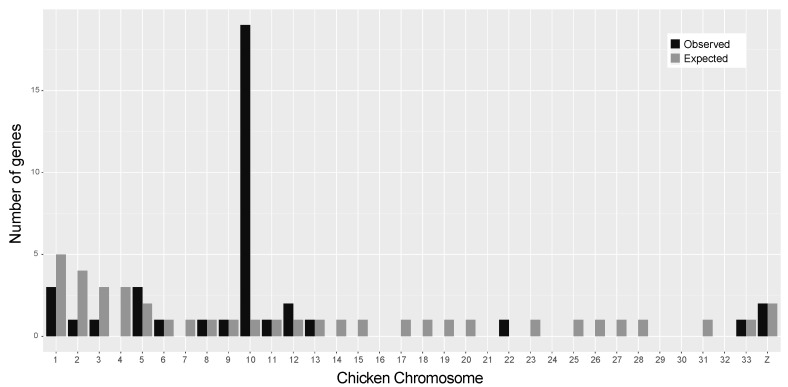
Localization of 38 chicken genes identified via BLAST of female-specific *C. pharbaungensis* RAD markers using *Hemidactylus turcicus* transcripts as an intermediary step. The number of observed (black) and expected (gray) genes with a BLAST match are shown for each chicken chromosome. The 19 genes found on chicken chromosome 10 are significantly more than expected (*p* = 5.525818 × 10^−21^), indicating synteny with *C. pharbaungensis* sex chromosomes. Plot made using ggplot2 and reshape2 in R v.3.6.3 [48,49,50].

**Figure 3 genes-12-00116-f003:**
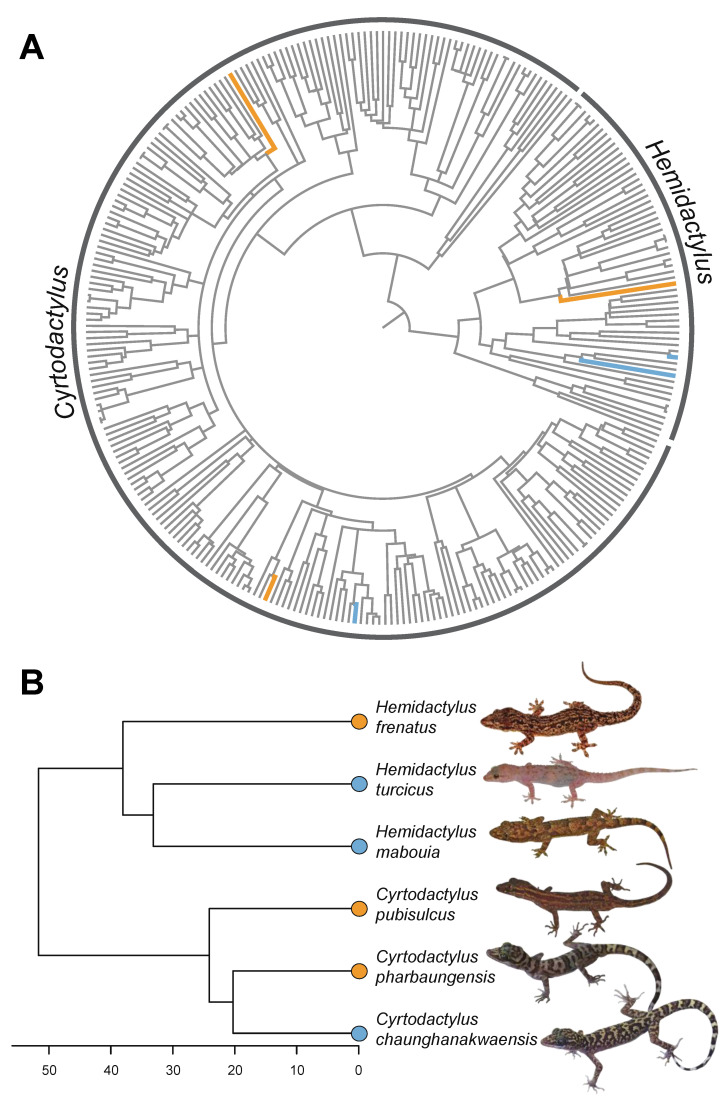
Phylogenetic placement of XX/XY and ZZ/ZW sex chromosomes in *Hemidactylus* and *Cyrtodactylus* geckos. (**A**) Maximum likelihood phylogeny of *Hemidactylus* and *Cyrtodactylus* geckos produced using the mitochondrial ND2 and adjacent tRNAs. Branch lengths are scaled in millions of years using secondary time calibrations. Species with known sex chromosome systems (see Figure 3B) are indicated with colored branches—orange branches indicate species with ZZ/ZW sex chromosomes, blue branches indicate species with XX/XY sex chromosomes. Taxa with gray branches are species with unknown sex chromosomes. (**B**) Phylogeny of species (pruned tree from Figure 3A) with known sex chromosome systems. Heterogamety is indicated at tips: blue for XX/XY and orange for ZZ/ZW. Time-scale in millions of years before present.

**Table 1 genes-12-00116-t001:** Newly designed PCR primers used to validate sex-specific RADseq markers.

Species	Primer Name	Sequence (5′ to 3′)
*Cyrtodactylus chaunghanakwaensis*	CyrtC_6-F	TCAGCCCTATATGCAACGGATC
	CyrtC_6-R	TGTCCCTTCAGTTGGTCCAAAA
	CyrtC_18-F	CCCGGTTAACTCTAGTCGCATT
	CyrtC-18-R	TGAGGGGTAGGCAAGATAAGGA
	CyrtC-151-F	CAGACTTGTCACTCACCCTGAA
	CyrtC_151-R	TGACTTCTCCTCATCTGGCAAC
*Cyrtodactylus pharbaungensis*	CyrtP_194-F	AATCAGGCGACCTTTAAGCTCA
	CyrtP_194-R	TGCAGACGTGATGTAAGGGAAA
	CyrtP_253-F	CTCAGTGGCTCCCTCGTTAATT
	CyrtP_253-R	TCTGTGTGGACTTTTTGGACCA

## Data Availability

The RADseq data are available at the NCBI Short Read Archive Bioproject PRJNA692216.

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
