# Peer review of "Sex Chromosome Turnover in Bent-Toed Geckos (Cyrtodactylus)"

_genes, 2021, doi:10.3390/genes12010116_

Round 1

Reviewer 1 Report

The present manuscript is another chapter in a series of works from Dr. Tony Gamble's lab, devoted to a study of sex chromosome evolution in gecko lizards. These lizards show high rates of sex chromosome evolution, with different systems occurring even in related species. This work presents another example of such plasticity, showing that two Cyrtodactylus species have different sex chromosome systems.

I did not find significant problems in the manuscript. My only question concerns Fig. 1, where the electrophoresis images are presented. On these images, "sex-specific" markers show bands also in the opposite sex (especially in A), although much less prominent. What is the reason behind this? This fact should be noted and explained in the Results section.

Author Response

We have added a sentence clarifying that small amounts of sequence similarity between sex chromosome alleles can produce faint bands in the homogametic sex.

Reviewer 2 Report

The proposed manuscript (ms) greatly contributes to the study of sex chromosome evolution and turnovers. I have not found any fundamental issues in proposed ms. The ms is technically sound, presented in an intelligible fashion and written in good-quality English. Clear and not complicated sentences make the ms easy and interesting to read.

I Have only a few minor suggestions:

1) The end of the first paragraph of introduction, it is not clearly indicated which fundamental evolutionary questions you would investigate. Try to specific this information.

2) I consider it is important to clearly describe origin of studied species at the beginning of M&M section. There is no information if animals originate from natural population or artificial crossing, no information about country (only Myanmar is mentioned in introduction) and exact locality where they were caught.

3) The CyrtP_253 marker showed PCR amplicons of different size in males and females (Figure 1). How it can be caused? Are there any other studies describing amplicons of different size between sexes? Any hypothesis would have deserved to be discussed.

4) Figure 1A and B are convincing but a ladder (as a proof of amplicon size) and a negative sample are missing. Could authors (i) put samples on the gel again or re-do PRC with negative sample and (ii) update the Figure 1?

5) In the discussion section, authors briefly mentioned an example of intermediate step(s) for BLAST searches in another study. Could they include more information, e.g., what kind of frog species? What species served for mapping?

6) There is the typo in reference number 32 (cncestry vs. ancestry?)

Author Response

1) We had added more detail to this paragraph:

"This discovery further highlights the extraordinary number of transitions among gekkotan sex chromosomes and the enormous gap in our knowledge surrounding the phylogenetic distribution of gecko sex chromosomes and evolutionary processes that lead to sex chromosome transitions."

2) We have added additional information on specimen collection.

"Cyrtodactylus chaunghanakwaensis [22] and C. pharbaungensis [23] are range-restricted endemics known only from small isolated karstic hills from Mon State in the Salween Basin of southern Myanmar and were collected from their type localities at Chaunghanakwa Hill and Pharbaung Cave (Supplemental Table 1)."

3) We have added a clarifying sentence to this paragraph.

"These PCR products correspond to amplification of the W allele in females and the Z allele in males Although the Z and W alleles have diverged to the point that they are different sizes, enough sequence similarity presumably exists that the primers designed for the W allele still amplify the Z allele in males. In females, the W allele is preferentially amplified over the Z allele, producing a single band." 

4) We have fixed the figure to include the negative control and ladder.

5) We have included more detail for this method.

"In that study, Rana arvalis sex-linked markers were iteratively mapped to a Rana temporaria draft genome. Those R. temporaria fragments were mapped to the Nanorana parkeri genome. Finally, the matching N. parkeri scaffolds were mapped to the highly contiguous Xenopus tropicalis genome."

6) We have fixed this typo.